# Endothelial Dysfunction, HMGB1, and Dengue: An Enigma to Solve

**DOI:** 10.3390/v14081765

**Published:** 2022-08-12

**Authors:** María-Angélica Calderón-Peláez, Carolina Coronel-Ruiz, Jaime E. Castellanos, Myriam L. Velandia-Romero

**Affiliations:** Universidad El Bosque, Vice-Chancellor of Research, Virology Group, Bogotá 110121, Colombia

**Keywords:** biomarkers, DENV, endothelial dysfunction, HMGB1, immune response

## Abstract

Dengue is a viral infection caused by dengue virus (DENV), which has a significant impact on public health worldwide. Although most infections are asymptomatic, a series of severe clinical manifestations such as hemorrhage and plasma leakage can occur during the severe presentation of the disease. This suggests that the virus or host immune response may affect the protective function of endothelial barriers, ultimately being considered the most relevant event in severe and fatal dengue pathogenesis. The mechanisms that induce these alterations are diverse. It has been suggested that the high mobility group box 1 protein (HMGB1) may be involved in endothelial dysfunction. This non-histone nuclear protein has different immunomodulatory activities and belongs to the alarmin group. High concentrations of HMGB1 have been detected in patients with several infectious diseases, including dengue, and it could be considered as a biomarker for the early diagnosis of dengue and a predictor of complications of the disease. This review summarizes the main features of dengue infection and describes the known causes associated with endothelial dysfunction, highlighting the involvement and possible relationship between HMGB1 and DENV.

## 1. Introduction

Viral infections are among the most important public health concerns. An example of this is the coronavirus disease 2019 (COVID-19) pandemic caused by severe acute respiratory syndrome coronavirus 2 (SARS-CoV-2), which claimed ~6.39 million lives worldwide, as of July 2022 [1]. Other non-pandemic viruses remain relevant, such as arboviruses (arthropod-borne viruses), which cause severe febrile syndromes with or without hemorrhage and can be disabling and leave sequelae or cause death in a considerable percentage of patients.

It is difficult to assess the global burden of arboviral diseases. Aspects such as poverty, limited access to basic public services (drinking water and waste collection), medical care, and difficulties in clinical and laboratory diagnosis can underestimate the rate of cases. Furthermore, low levels of education and lack of knowledge about the prevention and transmission of these diseases contribute to the spread of the virus [2,3]. The latter maintains an endless cycle of infection, particularly in endemic countries [4,5,6].

Among arboviruses, dengue virus (DENV) has the greatest global impact, with more than 390 million estimated annual cases worldwide [7,8] and at least 3900 million people at risk of infection every year [6]. Importantly, three-quarters of the total number of infected people are asymptomatic, and 96 million are considered symptomatic cases, with or without complications [7,8].

The dengue mortality rate ranges between 4% and 10% in Southeast Asia and the Americas [9,10]; however, this rate may also be underestimated, demonstrating the complexity and high burden of the disease [6]. Additionally, in endemic and hyperendemic countries, surveillance systems (procedures that allow assessment of the damage caused by the disease) have limitations for case confirmation and reporting, and there is often a disagreement between clinical diagnosis and laboratory test results. Consequently, many cases are febrile syndromes without complications [11].

Approximately 10% of dengue cases progress to severe forms of the disease, with the most frequent signs being plasma leakage, hemorrhage, or organ affection [12]. Endothelial dysfunction and vascular permeability are the key events of dengue disease and contribute to the efficient spread of DENV, immune cell migration, and release of soluble molecules from the blood to the tissue stroma, thus increasing disease severity [13,14,15].

These findings highlight the need to identify molecules that can be used as biomarkers to improve the diagnosis and prognosis of complicated dengue cases. However, to date, of the diverse molecules produced and released into the blood during DENV infection [16], no molecule has been defined or established as a dengue biomarker. Therefore, given the need to understand the pathogenesis of dengue and identify differential changes between molecules associated with disease severity, this review briefly describes DENV, infection, and some of the cellular and molecular events that may be related to endothelial dysfunction (ED). Here, we describe and highlight the possible role of the high mobility group box 1 (HMGB1) protein during the early response of the endothelium to DENV infection.

DENV is a flavivirus (Flaviviridae family) that is transmitted to humans by the bite of female mosquitoes of the genus *Aedes—A. aegyptii* or *A. albopictus* [17,18]. Its genome is a single strand of positive-sense RNA of approximately 11 kb, which codes for three structural proteins: the capsid (C), pre-membrane (prM), and envelope (E), and seven non-structural proteins (NS), NS1, NS2A, NS2B, NS3, NS4A, NS4B, and NS5 mainly involved in viral replication and assembly [15,17,19].

This virus circulates in tropical and subtropical regions and causes 390 million infections per year, with more than 500,000 hospitalizations [20] and more than 20,000 deaths [15,21,22]. The estimated burden of the disease (medical care, vector control and surveillance, productivity loss, and sudden deaths) is approximately $39 billion per year, of which $4 billion corresponds to disease expenses in the Americas [23].

In Colombia, the country with the second highest number of dengue cases in America, it was estimated that the total cost of the disease (including outpatient and hospital settings) has been increasing [24]. It has been reported that from 2013 to 2016, at least 1396 deaths were caused by DENV in Colombia. This study also estimated the burden of disease as 130,733 disability-adjusted life years (DALYs) [25], highlighting the complexity and high impact of the disease. However, these estimates assume constant annual infection rates and do not consider geographic or temporal variations in the incidence or symptomatic cases. Therefore, some authors have suggested that the actual number of DENV cases and deaths is much higher (up to three times more) [26].

DENV is antigenically classified into four serotypes, known as DENV 1–4, which circulate in all tropical and subtropical areas of the world [5,8,17,27]. These viruses can cause asymptomatic infection or febrile illness, with or without complications [20,28]. The clinical diagnosis of dengue is problematic, given that it depends mainly on the time of infection and unspecified signs and symptoms. For example, during the early phase of the disease, dengue can be confused with the common cold or present with symptoms similar to influenza, zika, chikungunya, yellow fever, or malaria infections, among others [29]. In addition, the signs and symptoms of the disease are dynamic and depend on viremia and the immune response related to the production of specific antibodies against DENV, which may vary depending on the patient’s infection history with DENV [30]. Approximately a quarter of patients develop severe forms of the disease, which are life-threatening and typically occur during the so-called severe dengue (SD) (previously called hemorrhagic dengue, DHF, and dengue shock syndrome, DSS). During SD, the most frequent clinical signs are plasmatic or capillary extravasation accompanied by thrombocytopenia, hemorrhage, impaired hemostasis, and organ damage in the liver, brain, and heart [12]. This suggests that increased vascular permeability is the main indicator of DENV infection and affects the endothelial barriers of organs and tissues [14], favoring the passage of cells and molecules from the blood to the stroma [13,14].

## 2. Endothelial Dysfunction: A Key Condition during DENV Pathogenesis

Endothelial cells (ECs) line the blood vessels and capillaries and form a highly selective physical barrier [31]. Owing to the importance of its tightly regulated functions, the vascular endothelium is a complex organ that can interact with and respond to its microenvironment in a controlled manner, moving from an inactive to an active state in a short time [32]. Among the essential components for the maintenance of ECs are the tight junction proteins (TJPs) and the junctional adhesion molecules (JAMs), which are responsible for maintaining low levels of transcytosis (passage of molecules through the cells, using vesicles) and paracellular diffusion (passage of molecules through the intercellular space of two adjacent cells) [33].

These ECs are central elements in the metabolism and physiology of tissues and organs and are the first to be affected during viral infections, which can induce cell death and, consequently, organ dysfunction [32]. In addition, ECs are susceptible to DENV infection, since post-mortem biopsies of SD cases have detected viral antigens, morphological changes, and capillary inflammation [34]. These changes are associated with monocyte diapedesis, edema, inflammation, and perivascular cell accumulation [35]. The susceptibility of these cells to the virus has also been reported using in vitro models with EC lines or primary cultures from the lung or spleen, where protein expression and efficient production of infectious viral particles have been reported [36,37]. However, this susceptibility to the virus does not explain the endothelial dysfunction (ED) or severe tissue events that occur during infection.

In this regard, the most frequent hypotheses associated with ED can be grouped into three areas: (1) direct infection of myeloid cells, in which cytokine production affects endothelial barriers, selection of viral strains of high or low pathogenicity [38], and functional disorders associated with single nucleotide polymorphisms (SNPs) in immune-response-related genes associated with susceptibility (e.g., IL-10, TNF-α, TGFβ, JAK1) or protection (e.g., FcγRIIa, VDR, TAP, MBL) [39,40]; (2) antibody-mediated enhanced infection (ADE), in which sub-neutralizing antibodies produced during previous infections interact with the FcγR receptors of immune cells, leading to virus opsonization and increased infection of different cells, including ECs [41,42]; and (3) an uncontrolled over-activated T cell response due to infection induces the release of high amounts of cytokines (cytokine storm) [43], which also induces the activation of the complement system in a heterotypic secondary infection, directly affecting ECs. Unfortunately, no adequate animal model is available to study dengue pathogenesis [43,44] or disease progression [15]. Figure 1 summarizes the major events and elements associated with ED produced during DENV.

Currently, some evidence shows that DENV activates ECs, which play a crucial role in the immune response to infection; therefore, they are considered markers of damage and dysfunction. For example, it has been observed that circulating ECs (CECs) are detected during the defervescence phase and are associated with an increase in the expression of soluble adhesion molecules, such as vascular cell adhesion molecule (VCAM) and intercellular cell adhesion molecule (ICAM). The latter results from the strong activation of the endothelial tissue (as a consequence of the infection), inducing the separation of the ECs and a loss of tissue integrity, explaining the extravasation of plasma during the acute phase of the disease [47]. Changes in fibrinolysis and coagulation have also been reported, such as increased levels of soluble thrombomodulin and von Willebrand factor (vWF), which are associated with disease severity and can be considered predictors of endothelial activation and disease progression [15].

Likewise, disruption of the endothelial glycocalyx (EGL) has been postulated to be a central factor in ED. The integrity of this structure is essential for tissue homeostasis [48,49]. However, in vitro models using ECs from human lung microvasculature and in vivo models using mouse ECs showed that the DENV NS1 protein (which normally circulates in the bloodstream during infection) induced the degradation of sialic acid and heparan sulfate proteoglycans (components of EGL) [48,50]. Therefore, the destruction of EGL and the release of other components, such as hyaluronic acid, heparan sulfate, and syndecan-1 could be evaluated in patient samples, and their concentration levels could be associated with disease severity and mortality of patients [51].

However, it is possible that factors not yet identified play important roles in the development of SD in patients with primary infection with “naive” or previously unstimulated immune systems [41]. Table 1 summarizes the main events and elements associated with ED due to DENV infection.

## 3. The HMGB1 Protein: A New Actor in the Pathogenesis

One of the molecules that could be associated with ED and the severity of dengue is the high mobility group 1 protein (HMGB1) [67], also known as amphoterin or HMG1. This 215 amino acid (30 kDa) protein is highly conserved in eukaryotic organisms [68].

HMGB1 is commonly located in the nucleus, where it is involved in DNA stability, repair, transcription, and recombination. This protein is organized into three main functional domains: box A (residues 1–79), box B (residues 89–163) with positive charge, both characterized as DNA binding domains, and an acidic tail that is negatively charged [69]. Importantly, domain box A has an anti-inflammatory effect, while box B constitutes the proinflammatory activity of the protein; the localization of this residue in HMGB1 is associated with its binding and interaction ability, which induce its activities and functions [70].

During the non-pathological state this protein remains in the cell nucleus; however, under certain stimuli, it can be transported to the cytoplasm, where it undergoes a series of post-translational modifications, such as acetylation, phosphorylation, methylation, glycosylation, oxidation, and ADP ribosylation, which modulate its structure and subsequent function [71].

Acetylation of HMGB1 is a reversible and dynamic modification that occurs at the N-terminus (residues Lys2 and Lys11) by the activity of histone acetyltransferases (HATs) and/or histone deacetylases (HDACs) and it is decisive for protein mobilization from the nucleus to the cytoplasm and affects its secretion [72]. Methylation at the Lys42 protein residue induces conformational changes that reduce its ability to bind to DNA, thereby increasing its release from the nucleus via passive diffusion. This modification is mainly detected in the cytoplasm of neutrophils and other immune cells [73]. In contrast, phosphorylation occurs mainly at residues Ser34, Ser38, Ser41, Ser45, Ser52, and Ser180 and is mediated by calcium/calmodulin-dependent kinase (CaMK), among others. This modification allows the protein to be exported from the nucleus, transported to the cytoplasm, and subsequently secreted into the extracellular medium [74].

HMGB1 also undergoes oxidation at Cys23, Cys45, and Cys106 residues, which induces conformational changes under oxidative stress conditions (in which reactive oxygen species and different signaling pathways interact), generating protein translocation from the nucleus to the cytoplasm. Some reports have shown that the oxidation of Cys106 is associated with the inhibition of the inflammatory activity of HMGB1. Likewise, oxidation also induces HMGB1 release and its activity as a pro-inflammatory protein in immune response events [73]. Similarly, glycosylation is associated with translocation and extracellular secretion of HMGB1. N-glycosylation has been identified at the Asn37, Asn134, and Asn135 residues; the latter is related to extracellular protein activity [73]. Finally, ADP ribosylation of HMGB1 (poly ADP-ribosylation) negatively regulates gene transcription and leads to protein translocation from the nucleus and its subsequent release during cell death, mainly in necrosis [74].

Depending on the stimuli, the protein may have one or more of the modifications previously mentioned and have pleiotropic activity. In the cytoplasm, HMGB1 acts as a chaperone-type protein, inhibiting the aggregation of polyglutamine to beclin-1 (associated with autophagy), which enables the regulation of this process and cell death [75,76]. It has also been described that the regulated accumulation of HMGB1 in the cytoplasm is determined by the JAK/STAT1 signaling pathway, which induces protein translocation and the formation of disulfide bonds (which exert an inflammatory response through TLR4 and the factor of myeloid differentiation, MD2) [72]. Additionally, HMGB1 acts as a sentinel by detecting nucleic acids, activating the signaling pathways involved in the innate immune response, and inducing the production of proinflammatory cytokines [77].

Interestingly, HMGB1 has other functions outside the cell [75], acting as an alarmin or damage-associated molecular pattern (DAMP). HMGB1 is recognized by the receptor for advanced glycation end products (RAGE) and Toll-like receptors (TLRs) 2, 4, and 9 [71,78]. This protein triggers an inflammatory response by attracting cells from the immune system, promoting their activation and proliferation, and inducing tissue repair in response to damage. Likewise, HMGB1 can modify the immune response, promoting immunity or tolerance, which is reflected in the activation of effector T cells and the suppression of regulatory T cells [79].

This demonstrates the pleiotropic activity of HMGB1. As a cytokine, it is involved in signal transduction and coordinates cellular activities by interacting with other proteins, such as pathogen recognition receptors (PRR), the expression of which differs depending on the cell type. For example, in NK cells, helper, and regulatory T cells, the activation of RAGE, TLR2/TLR4, and TIM3 has been demonstrated, while in B cells, TLR9 is also activated [80,81]

Finally, HMGB1 must be released into the bloodstream to perform its immune function. In immune cells, it has been reported that this process occurs in two steps: (I) HMGB1 acetylation is induced, causing its translocation from the nucleus to the cytoplasm and its accumulation; and (II) secretion of the protein occurs, but its release mechanism varies depending on the damage signals that originate during apoptosis (Figure 2).

## 4. HMGB1 and ED: What Other Diseases Teach Us, and What We Know about DENV

The first report linking the HMGB1 protein to infection and disease progression was in sepsis [82]. As HMGB1 maintains and prolongs the pathological process of sepsis, different studies have focused on describing the role of this protein in this condition. Thus, HMGB1 has been found in patients’ sera from the first 24 h of symptom onset until 96 h, when signs associated with disease severity are identified. The effect of HMGB1 concentration on the permeability of human umbilical vein endothelial cells (HUVEC) seeded on semi-permeable membranes and exposed to patient serum samples showed a permeability increase from 18 h post-stimulation (hps) to 48 hps. Interestingly, treatment of sera with ethyl pyruvate (EP, an inhibitor of HMGB1) significantly reduced endothelial permeability, demonstrating that HMGB1 is one of the mediators of early detection of endothelial dysfunction during the infectious process [83].

Similar results have been reported during infection with influenza A H1N1 virus: HMGB1 is released by necrotic cells, inducing an exacerbated production of cytokines and inflammatory mediators such as IL-1ß, TNF-α, IL-6, and IL-23, and promoting alteration of the blood–brain barrier (BBB) [84]. These authors also reported that HMGB1 levels in sera, but not in the cerebrospinal fluid (CSF), of patients diagnosed with virus-associated encephalopathy and who developed neurological symptoms were higher than those observed in infected individuals with a diagnosis of encephalopathy without neurological sequelae (median 17.4 ng/mL vs. 6.8 ng/mL). These results suggest that, under specific conditions of the neuropathology of the infection, HMGB1 has a differential dynamic [84].

Another study reported by Rus et al. (2016) quantified HMGB1 levels in sera from patients infected with hemorrhagic viruses such as Hantavirus, Dobrava-Belgrade virus (DOBV), Puumala virus (PUUV), Crimean-Congo hemorrhagic fever (CCHF), and hemorrhagic fever with renal syndrome (HFRS). The study found that only patients infected with HCCF, DOBV, and PUUV presented elevated levels of HMGB1 compared with healthy volunteers (*p* < 0.0001). Additionally, HMGB1 levels increase in patients with severe disease. Again, these findings highlight the importance of considering HMGB1 as a possible biomarker of disease severity during hemorrhagic infections [85].

In contrast, during hepatitis B virus (HBV) infection, the release of HMGB1 (as a DAMP pattern), along with other cytokines, contributes to chronic liver inflammation [86]. This finding was observed in murine models of acute and chronic liver failure (ACLF), where severe cases of HBV infection were associated with an increase in HMGB1 transcripts and proteins. In murine models, HMGB1 acts as a late proinflammatory mediator involved in mice lethality. The treatment of mice with neutralizing antibodies (specific for HMGB1) protected the liver, prevented liver failure, and improved the survival rate of the animals. These results suggest that HMGB1 contributes to persistent liver inflammation, which is considered a critical mediator of lethality during ACLF processes [78,87].

Similarly, significant differences in HMGB1 levels were found in patients with acute self-limiting hepatitis E virus (HEV) infection (112.6 ng/mL) and ACLF (225 ng/mL) compared to healthy volunteers (12.4 ng/mL). Furthermore, a positive correlation was observed between circulating HMGB1 levels and its transcript, indicating that hepatocyte damage during infection is directly related to high levels of circulating HMGB1, suggesting that this protein is a possible marker of liver failure [88].

Interestingly, the use of HMGB1 as a prognostic marker of disease has also been reported in patients with human immunodeficiency virus (HIV) since increased serum levels were found in patients with neurological alterations. In addition, an inverse correlation between HMGB1 levels and neutralizing antibodies has been reported, suggesting that these antibodies modulate HMGB1 proinflammatory activity [89].

Regarding DENV, Chen et al. (2008) reported the first evidence linking HMGB1 to DENV infection. Using adenocarcinoma human alveolar basal epithelial cells (A549), they evaluated the effect of DENV-2 infection (at a multiplicity of infection [MOI] of 10) on survival, plasma membrane integrity, and expression of genes associated with apoptosis. The results showed a cytopathic effect 24 h post-infection (hpi), coinciding with the viral production peak. In addition, they observed that between 24 and 48 hpi, the virus induced the loss of cell viability associated with apoptosis, as evidenced by Mcl-1, Bax, and Bcl-xL gene expression and significant release of lactate dehydrogenase (LDH). Interestingly, HMGB1 was detected in cell supernatants at 72 hpi, suggesting that infection induced a passive release of HMGB1 by necrotic cells, which could be a factor that enhances the inflammation observed during DENV disease [90].

Kamau et al. (2009) investigated HMGB1 release from immature dendritic cells (DC) and human CD3+ T cells after infection with DENV-2. Under these conditions, the authors evaluated the translocation of HMGB1 from the nucleus to the cytoplasm and extracellular release at 24 hpi using confocal microscopy and ELISA. This effect increased when the cells were exposed to higher MOIs (10 and 100), indicating that protein translocation and subsequent release are viral-load-dependent processes. Additionally, they reported that HMGB1 plays an essential role in the regulation of cytokine production and induces the secretion of TNF-α, IL-6, IL-8, and IFN-α in infected DCs. Based on these results, the authors proposed that DENV-2 infection induced cytokine expression of other proteins, such as HMGB1, which interact in a complex manner, demonstrating its participation in a cytokine storm [91].

Allonso et al. (2012) evaluated HMGB1 levels in the sera of adult patients with primary or secondary DENV infection (116 samples) compared to healthy volunteers (33 samples). Patients were classified in three post-infection periods according to the days of symptoms onset in very early (0–3 days), early (4–7 days) and late (14–30 days). The results showed a significant increase in the concentration of this protein in infected patients compared with other febrile patients (not DENV-infected). Additionally, higher HMGB-1 levels were observed in the very early stages of the disease, with no differences between primary and secondary infections, although levels were slightly higher in the latter. These results are of great importance because this study is the first to show that HMGB1 can be elevated in DENV-infected patients, especially during the first days of illness, suggesting that HMGB1 levels could be associated with symptoms [92].

These findings suggest that HMGB1 is a potential biomarker for early detection and diagnosis of dengue cases. Therefore, the same authors evaluated serum HMGB1 levels in 205 samples from adult individuals with dengue-like fever (DLF), non-laboratory-confirmed clinical cases of dengue (CC), and healthy blood donors as the negative control. Infected samples were divided into three groups according to the days of symptoms onset and the type of infection (primary or secondary), as previously mentioned. Using capture ELISA, the study showed low levels of HMGB1 in the sera from healthy donors, and only 12.5% of DLF sera were positive. However, there were significant changes in HMGB1 concentrations in DENV patients compared with DLFs; while in the CC, the protein presented values similar to those of the infected patients. These results indicate that HMGB1 is a differential marker of dengue cases, which reinforces the potential of this protein as a biomarker on the first days of the disease [93], suggesting that, as in other infectious events, HMGB1 appears to play a role in the development and establishment of DENV.

Based on the evidence previously described, it is suggested that the HMGB1 may also play an important role in the development of severe forms of DENV disease (given that this is also considered a hemorrhagic virus). However, evidence to date is still limited, and the cells involved in HMGB1 release during DENV infection have not yet been evaluated.

Ong et al. (2012) evaluated the translocation and release of HMGB1 in DENV-infected peripheral blood mononuclear cells (PBMC) and K562 erythro-leukemia cells at different MOIs (1 or 10). In both cells, the authors reported translocation from the nucleus to the cytoplasm and its secretion into the extracellular medium, a phenomenon inhibited by treatment with ethyl pyruvate, without affecting the replication or release of viral particles (as observed in plaque assays). Additionally, the inoculation of human recombinant HMGB1 protein (hrHMGB1) or DENV-infected K562 supernatants on HUVEC of an endothelial in vitro model caused a decrease in the values of transendothelial resistance (TEER) and increased permeability. These changes were also inhibited by EP treatment. These findings allowed the authors to conclude that DENV infection induced HMGB1 mobilization and release, which may disrupt the integrity of EC, explaining the increased vascular permeability observed in patients with SD, which is one of the critical events in dengue pathogenesis [94].

Zou et al. (2015) attempted to explain the mechanism by which HMGB1 modifies endothelial physiology. For this, they seeded the EC line EAhy.926 on semi-permeable membranes that were exposed to different concentrations of hrHMGB1. Concentrations greater than or equal to 200 ng/mL of the protein caused a decrease in TEER from 6 h post-stimulus (hps) without causing damage to cell viability. These changes were related to a reduction in the expression of the junctional adhesion molecule VE-cadherin (one of the proteins responsible for maintaining endothelial integrity), the formation of interendothelial gaps, and the activation and phosphorylation of Src protein, previously associated with endothelial hyperpermeability. The activation of Src seems to be regulated by the “store-operated calcium channel” (SOCE), suggesting an increase in intracellular calcium induces VE-cadherin reorganization. These results strongly suggest a new mechanism that modulates vascular permeability, where HMGB1 activates SOCE-dependent calcium entry and Src activation, triggering changes in intercellular binding proteins and, subsequently, endothelial hyperpermeability [95]. These data highlight the role of HMGB1 in EC activation and induction of endothelial permeability during the pathogenesis of DENV disease.

Given the interest in decreasing or reversing the effects of HMGB1 on the endothelium during DENV infection, Zianal et al. (2017), evaluated the antiviral activity of resveratrol (RESV), a known anti-inflammatory and antioxidant agent, and its effect on HMGB1 translocation and release in human hepatocarcinoma (Huh7) cells infected with DENV-2 (MOI 1). The data showed that infected cells treated with different concentrations of RESV significantly decreased viral titers (plaque assays) and NS3 viral protein expression (Western blot assays). Significantly, RESV treatment lead to protein deacetylation (via Sirt-1), restricting HMGB1 translocation from the nucleus to the cytoplasm, and in consequence, decreasing the amount of HMGB1 in the supernatants, In contrast, HMGB1 or Sirt-1 knockdown assays demonstrated the importance of the protein in the antiviral response because the lack of HMGB1 in the nucleus enhanced viral replication, where Sirt-1 coordinates the efficient antiviral response. These findings demonstrate the importance of nuclear HMGB1 in establishing an efficient antiviral response that can be regulated by RESV treatment through inhibiting HMGB-1 translocation and DENV replication [96].

Sirt-1 activation by RESV is associated with decreased expression of HMGB1, MyD88, and NF-kB mRNA levels [97]. It also downregulates the expression of TLR4 and the dependent signaling pathways mediated by it. Recently, Chaudhary et al. (2022) reported that RESV antagonizes DENV replication by inhibiting the cytoplasmic translocation of HMGB1 and its accumulation in the nucleus, where it induces the activation of interferon-stimulated genes (ISGs) [98]. These results are consistent with the data obtained by Zaitnal et al. (2017) on the antiviral response of RESV in dengue infection.

A recent study showed that infection of A549 cells with DENV-2 for 24 h (MOI of 5) induced HMGB1 accumulation in the cytoplasm and its subsequent release. This study showed that treating cells with EP increased the transcription of viral NS5 and capsid proteins, inhibiting HMGB1 release in infected cells. In a complementary way, the silencing of HMGB1 in these cells resulted in a reduction in viral RNA (~75%) and infectious viral particles according to titration assays. Likewise, a decrease in the mRNA expression of TNF-α, IL-6, IL1β, NF-kB, and IFN-β was observed, indicating that HMGB1 has a pro-viral effect on infected cells, mediated by proinflammatory cytokine expression and dependent on NF-kB activity, allowing viral replication and spreading [99]. Figure 3 summarizes the above-mentioned effect of HMGB1 in endothelial cells with and without DENV infection.

During DENV infection (red), endothelial (as well as immune) cells can release high concentrations of HMGB1. Viral replication and the accumulation of viral RNA and proteins (such as C and NS5) in the cytoplasm seems to favor HMGB1 translocation from N and its subsequent release to the extracellular medium (EM). When translocated to the cytoplasm, HMGB1 favors the release of the NF-kB from IKK, allowing it to go into N, where it induces the activation of genes associated with the immune response. HMGB1 translocation also induces cell death (apoptosis and/or necrosis) by different pathways, favors Src phosphorylation, and increases the concentration of calcium that induces the degradation of the adhesion protein VE-CAD. Finally, the release of cytokines, HMGB1, and viral particles into the EM affects the integrity of the endothelial barrier, leading to tissue hyperpermeability. The variation in the concentration of HMGB1 in the N, cytoplasm, or EM, in each condition (infection and non-infection) is represented as a concentration gradient (grey triangle)

Finally, circulation of HMGB1 during other flavivirus infections has been reported. For example, it was shown that during West Nile virus (WNV) encephalitis, serum levels of HMGB1 were higher than those in individuals with WNV fever, corroborating the role of this protein as a predictive biomarker for the prognosis and differentiation of WNV patients [100]. Given that HMGB1 has been detected during other viral diseases, Table 2 summarizes the involvement of HMGB1 in different viral infections, highlighting the possible role of the protein as a possible biomarker of disease prognosis.

These findings highlighted the HMGB1 protein as an alarmin released by cells immediately after an aggressive stimulus, which triggers the activation of alerting neighboring cells and induces different types of responses. However, despite these results, the production of HMGB1 during DENV infection and its effect on disease pathogenesis remains poorly understood.

## 5. Conclusions

Endothelial damage is a critical indicator of disease severity and potential lethality. Several mechanisms have been postulated wherein ECs are thought to be affected during DENV infection, where the uncontrolled reaction of the immune system and the elevated concentrations of some cytokines seem essential for endothelial damage. However, other inflammatory mediators could be involved in the disruption of this tissue, including the HMGB1 protein, which, with a dual nature and pleiotropic activity depending on its cellular localization, can be found intranuclearly, in the cytoplasm, or be released into the extracellular medium, where it acts as a proinflammatory mediator.

The presence of HMGB1 in the serum has been reported in patients with different diseases such as sepsis, cancer, and viral infections (e.g., HBV, HBE, HIV, WNV) as well as hemorrhagic-type viruses, including DOBV, where it has been proposed as a possible biomarker of disease progression in severe cases. Importantly, this work described the findings known during DENV infection, showing that this virus not only induces the expression and mobilization of HMGB1 in different cell models in a dose-dependent manner but it has also been detected in infected patient sera, especially in late infection periods. The effect of HMGB1 on different endothelial barrier models was detailed, showing that that protein induces an increase in endothelial permeability through distinct signaling pathways without the need for DENV replication, which affects protein release or the induced damage of the endothelial barrier. It seems that changes in protein location and mobilization depend on virus entry into cells.

The findings presented in this review support the hypothesis that HMGB1 is involved in DENV pathogenesis and could be a potential marker for diagnosing and managing dengue cases. However, further clinical and experimental evidence is needed to elucidate its role in endothelial dysfunction, as well as its potential as a diagnostic and prognostic tool for dengue among patients of different ages and infection conditions.

## Figures and Tables

**Figure 1 viruses-14-01765-f001:**
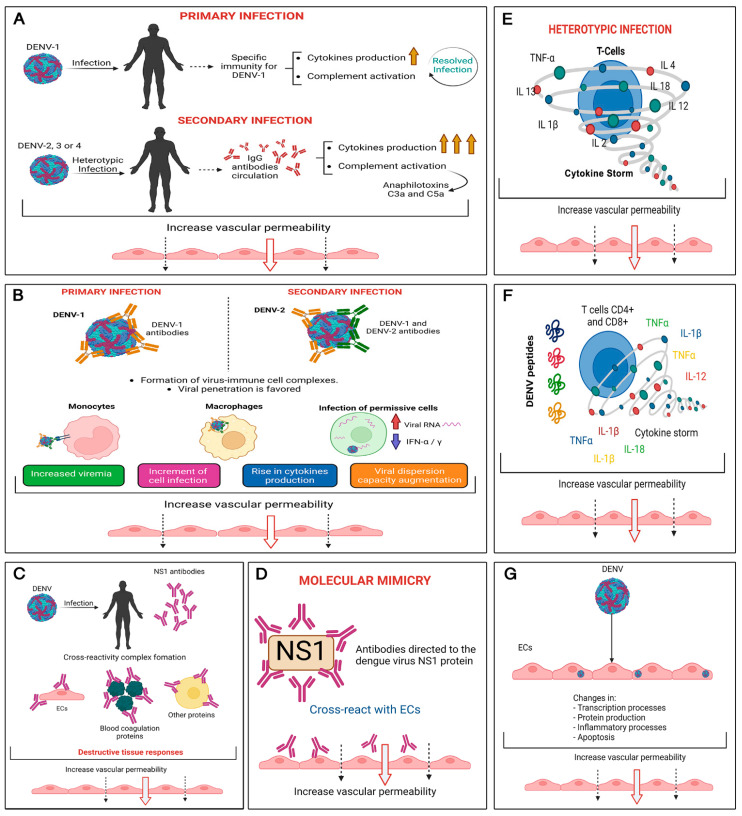
Molecular mechanisms associated with ED due to DENV. (**A**) Vascular permeability mediated by antigen (virus) + antibody (anti-virus) + complement proteins: in secondary heterotypic infection, the circulation of DENV-specific IgG antibodies activates the production of anaphylatoxins C3a and C5a of the complement system that mediate vascular permeability. (**B**) Antibody-mediated potentiation of the infection: in a secondary heterotypic infection (SHI), the antibodies from primary infection recognize and form complexes with the DENV serotype involved the secondary infection (reinfection). These complexes bind to monocytes and macrophages through Fc receptors that favors the viral entry, increasing the number of infected cells, viremia, and the cell-to-cell viral spread capacity that affects endothelial permeability. In addition, direct infection of permissive cells causes an intrinsic antibody-dependent enhancement (iADE) where viral replication and the consecutive production of viral RNA generate a downregulation on interferon alpha and gamma (IFN α/γ), and an upregulation of different cytokines that can be directly related to endothelial damage. (**C**,**D**) Heterophilic immunity and molecular mimicry: during DENV infection, antibodies directed against the NS1 protein are generated; these antibodies have cross-reactivity with ECs, blood coagulation proteins, and other cells such as hepatocytes. These antibodies, considered as mimetics, reach a pathogenic potential during a SHI, generating destructive responses in the endothelium. (**E**) Exaggerated T cell response: during SHI, different memory T cells release exaggerated amounts of cytokines that affect vascular permeability. (**F**) T cell response, guided by antigenic sin (tendency of immune system to preferentially use immune memory based on a previous infection when a second, slightly different version of this foreign entity is encountered): different variants of DENV peptides induce the production of different sets of cytokines. The response of T cells influenced by antigenic sin causes the uncontrolled cytokine production (cytokine storm), which increases vascular permeability. (**G**) Direct infection of ECs: DENV can directly infect ECs, triggering processes such as apoptosis and generating products that increase vascular permeability [38,45,46].

**Figure 2 viruses-14-01765-f002:**
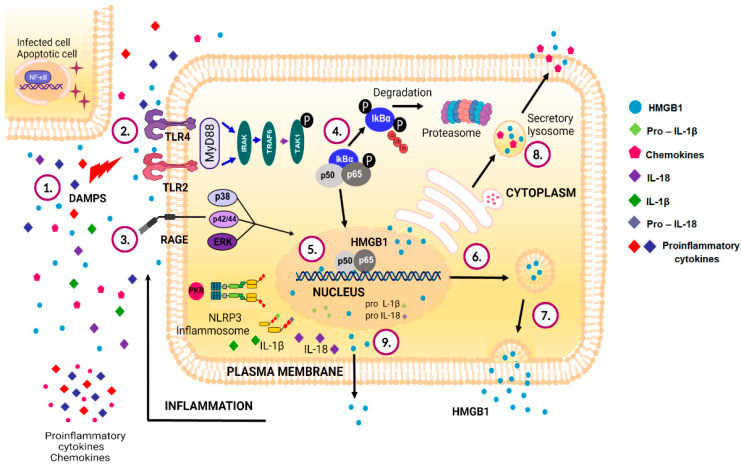
Mechanisms of intracellular mobilization and release of the HMGB1 protein. 1. HMGB1 protein and some molecules are released by cells, acting as alarmins (DAMPs) during apoptosis or necrosis, inducing cell activation and releasing proinflammatory molecules. 2. HMGB1 protein interaction with Toll-like receptors (TLR2 and TLR4) leads to the activation of MyD88-dependent signaling pathways. 3. Binding of HMGB1 to RAGE induces the activation of signaling pathways mediated by p38 protein, p42/44 complexes, and ERK family proteins, inducing the activation of transcription factors. 4. NF-kB activation is induced by the interaction between HMGB1 and its receptors. 5. HMGB1 protein in the nucleus acts as a DNA-binding protein and as a regulatory protein during replication, transcription, and DNA repair. 6. Cell activation induces HMGB1 translocation from the nucleus to the cytoplasm. 7. In immune cells, HMGB1 is transported from the cytoplasm to the plasma membrane during the release of microvesicles. 8. HMGB1 is transported into the cytoplasm and released into the extracellular medium through secretory lysosomes (unconventional secretion). 9. Cell activation mediated by PAMPs or DAMPs induces inflammasome activation, PKR phosphorylation, and HMGB1 translocation and release into the extracellular medium by pyroptosis.

**Figure 3 viruses-14-01765-f003:**
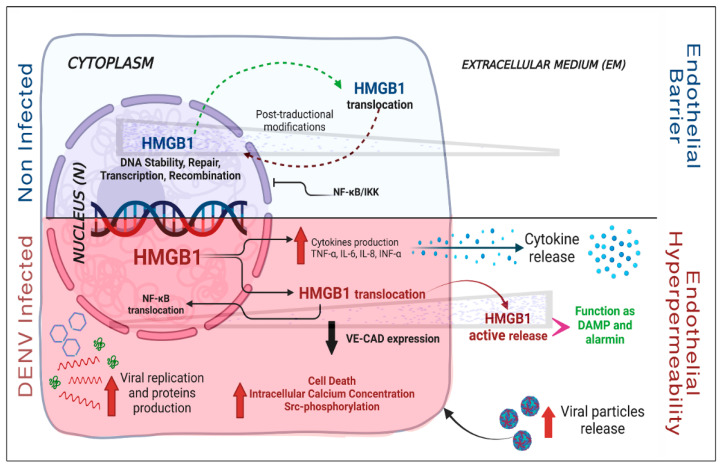
Evidence of HMGB1 protein localization and function in ECs during DENV infection. Without infection stimuli (blue), HMGB1 is mainly confined to the nucleus (N). In this condition, the gene transcription factor NF-kB remains in the cytoplasm inhibited by IKK. These mechanisms and others guarantee the endothelial barrier function.

**Table 1 viruses-14-01765-t001:** Events and molecules associated with ED as consequence of DENV infection.

Type of Marker	Event	Compared to DF	Population	Severity Biomarker?	Other Results	References
**Cells**	Number and circulation frequency of mDC and pDC	**↓**	Children (6 mo to 14 yo) and adults (>15 yo)	Yes, in children	Non mentioned	[52,53]
	Circulation frequency of NK and CD8+ T lymphocytes	**↓**	Adults (>14 yo)	No	↑ CD69, TIA-1, IL-15, CD38, CD44, LFA-1, CD11a, CD18	[54]
	Activation of cross-reactive T cells	**↑**	Children	Yes	↑ IFN-γ, TNF-α, IL-1, IL-6, IL-8, IL-10, CCL2 (MCP-1) and RANTES	[55]
	Platelet count	**↓**	Adults	Yes	Only if IL-10 is detected during the first days of the disease	[56,57]
			(>15 yo)			
**Cytokines**	IL-10	**↑**	Adults (>18 yo)	Yes	IL-10 potential marker for DF.	[58]
					↓ CD106, CD154, IL-4 e IL-33.	
	CD 121b	**↑**		Yes	Potential predictor of severity signs along with CD62E, CD62P, CD106 and IL-6	
	CD154	**↓**		No	↓ IL-4 and IFN-γ as the disease worsens	
	MIF	**↓**	Adults	Yes	↑ IL-6 e IL-10 and ↓ IFN-γ	[59]
**Chemokines**	CXCL-10	**↑**	Children	Yes	No differences between age groups were observed in protein levels	[60]
			(<15 yo)			
			Adults			
			(>15 yo)			
**Proteases**	Tryptase	**↑**	Children (6 mo to 15 yo)	Yes	↑ VEGF y ↓ sVEGFR-2 in DHF and DSS.	[61]
	Chymase	**↑**			↑ Chymase in DSS, ↔ between DF and DHF.	
					These changes are only observed in the first days of the disease	
**Proteins**	CRP	**↑**	Adults	Yes	↑ HA in acute phase of DHF and during DSS	[62]
			(>18 yo)			
	NS1	**↑**	Adults	Yes	NS1 detected after the fifth dpi, suggests bigger probability of DHF (OD 3.0)	[63]
			(>18 yo)			
	Soluble ST2	**↑**	Children	Yes	↑ ST2 soluble, TNF-a, IL-8 and IL-10 in patients with DHF compared to DF patients.	[64]
			(1 mo to 18 yo)			
			Adults			
			(>18 yo)			
**Others**	HA	**↑**	Children	No	↑ HA in patients with DHF and ↑↑ in patients with DSS	[65]
			(< 15 yo)			
			Adults	No	↑ During secondary infections. ↔ Between DF and DHF or DSS	[66]
			(>18 yo)			
	HS	**↑**	Adults	No	↑ Undifferentiated in patients with dengue	
			(>18 yo)			
	NO	**↑**	Adults	Yes	Significative differences between patients with DF and DHF	
			(>18 yo)			

Months old (mo); years old (yo); dengue fever (DF); dengue hemorrhagic fever (DHF); dengue shock syndrome (DSS); myeloid dendritic cells (mDC) and plasmacytoid dendritic cells (pDC); natural killer cells (NK); C-reactive protein (CRP); hyaluronic acid (HA); heparan sulfate (HS); nitric oxide (NO); vascular endothelial growth factor (VEGF); optic density (OD); soluble vascular endothelial growth factor 2 receptor (sVEGFR-2); days post-infection (dpi); ↑ rise; ↑↑ bigger rise; ↓ decrease; and ↔ no difference.

**Table 2 viruses-14-01765-t002:** HMGB1 in other viral infections.

Virus	Target (Tissue or Organ)	Type of Study	Description		HMGB1 ¿Biomarker?	REF
			Sample/Tissue	Results		
**HIV**	Systemic (≠organs)	**C.C:** 28 patients HIV + (21 Progressors; 7 LTNP). 25 IBD ≥ 18 yo	Plasma	**Both:** ↑ HMGB1 HIV+ and IBD vs. HC. **Only HIV+:** ↑MD-2	ND	[100]
	CNS	**C.C:** 103 patients (30 HAND; 73 No HAND) ≥ 18 yo	CSF/Blood	**CSF:** ↑ HMGB1 and ↑↑ ⍺-HMGB1. Iin addition, ↑↑IP-10 and MCP-1. Correlated with viral load	ND	[89]
	Systemic (≠ organs)	**C.C:** 30 patients (before and after c-ART) ≥ 18 yo	Blood	↔ Glu, Chol and HMGB1 before and after c-ART. After c-ART: ↑↑ CRP and Trig	ND	[101]
	Systemic (≠ organs)	**C.C:** 34 patients ≥ 18 yo (before and after c-ART). **In vivo:** 9 Pigtail Mq; 11 Rhesus Mq, Acute (29 dpi); chronic (197 dpi) SIV infected. (before and after c-ART).	Blood (PBMCs)/Plasma	**Both cond:** ↑↑ sCD14. **Mq:** ↑↑ HMGB1 (Bef c-ART, Acute and Chronic inf). **Hm:** ↔ HMGB1 Bef and After c-ART	ND	[102]
	Systemic (≠ organs)	**C.C:** 26 Pregnant HIV + 31 Pregnant HIV-2 HC. Women ≥ 18 yo	Serum/Cotyledons (placenta)/Decidua and Villi	**mRNA HMGB1:** ↑↑↑ Villi; ↑↑ Decidua. **Prot HMGB1:** ↓ Villi, ↓↓ Decidua. **Serum:** HMGB1 ↓↓	ND	[103]
	Systemic (≠ organs)	**C.C:** 39 patients (13 SID; 13 MID;13 HC) ≥ 18 yo	Blood (PBMCs)	↑↑↑ HMGB1 in SID ↑↑ HMGB1 in MID	ND	[104]
**CVB3**	Heart	**In vivo:** Male BALB/c mice (5 wo)	Heart	↑↑↑ HMGB1; ↑↑↑ TNF-α; ↑↑↑ IL1b; ↑↑↑ IL-10	ND	[105]
**HTLV-1**	T-Cells	**In vitro:** Cell lines: MT-2, MT-4, C5/MJ, SLB-1, HuT-102, MT-1, TL-OmI, ED-40515(-)		↑↑↑ HMGB1: All cells lines.	ND	[106]
**RV**	Liver Murine Biliary Atresia	**In vivo:** Tlr22/2 B6 mice, Tlr42/2 B10 mice	Liver/Cholangiocytes or Mø/Hepatic NK cells	↑ Expression HMGB1 in bile ducts and periductal area both in human and murine BA. Infection induces release of HMGB1 from cholangiocytes and Mø. HMGB1 induces activation of NK cells via TLR2 and TLR4	ND	[107]
	Liver and extrahepatic bile duct. Biliary Atresia	**In vivo:** Liver extrahepatic bile duct. **In vitro:** Mouse cholangiocyte cell line		↑ HMGB1 in serum of pups, 7 dpi with WT-RRV, ↑HMGB1 in supernatants from cells infected with WT-RRV, RRVVP4-K187R, and RRVVP4-D308A	ND	[108]
**HBV**	Liver Related acute-on-chronic liver failure	**C.C:** 50 patients with ACLF, 35 patients with LC, 35 with CHB and 35 with HS	Serum	↑ HMGB1 levels ↑ AUC values for HMMGB1	Yes	[109]
	Liver fibrosis	**C.C:** 189 CHB patients and 51 HC (>18 yo)	Serum/liver	↑ HMGB1 in patients from various fibrosis stages than control.	ND	[110]
	Hepatocellular carcinoma	**C.C:** Patients infected by HBV. **Ex vivo:** Tissue specimens (Liver). **In vitro:** Huh7 cells		HMGB1 expression is associated with the HCC pathological grade and the overall survival of patients	ND	[111]
	Hepatocellular carcinoma	**In vivo:** Mice C57BL/6J. **In vitro:** Huh7, HepG2, L02, HepG2.2.15 cells		HBx bound to HMGB1 in the cytoplasm, which triggered autophagy in hepatocytes	ND	[112]
	Primary Liver Cancer	**C.C:** 76 patients with primary liver cancer. **In vivo:** Male BALB/c nu/nu mice (6–8 wo), **Ex vivo:** Hm Liver resection of tumor samples, **In vitro:** HepG2, Huh7 cells.		HMGB1 expression and HBV infection status are positively related to PVTT in primary liver cancer	ND	[113]
**HBV/HCV**	Cirrohsis, Chronic liver disease	**C.C:** 160 patients with advanced-stage liver disease (≥18 yo)	Serum	No differences in serum marker concentration were found cirrhosis between HBV and HCV	ND	[114]
**HCV**	Liver	**In vitro:** Huh7	Huh7	HMGB1 negatively modulates HCV replication in the replicon system	ND	[115]
**FLUBV/WV/FLUAV/RSV/HRV**	Lung	**In vivo:** Cotton rats (4 to 6 wo), male or female	Serum/Lung	↑↑↑ HMGB1 in FLUAVA and FLUAB. ↑↑↑ HMGB1 in RSV ↑HMGB1 in HRV14 and HRV16 infection	ND	[116]
**FLUAV (H3N2)**	Lung	**In vitro:** Primary HMVECs cells		↑ HMGB1 at 12 h, 24 h of TNF-α stimulation. HMGB1 translocation	ND	[117]
**SARS-CoV-2**	Lung	67 critically ill COVID-19 patients from ICU	Plasma	↑ HMGB1 levels	Yes, for fatal outcome	[118]
		Patients (adults) with skin manifestations	Biopsies	↑↑↑ HMGB1 expression	ND	[119]
		Adults (N = 93). COVID-19 patients (23 admited to ICU, 19 outpatients mild moderate symptoms, 17 history COVID-19 infection) and 34 HC	Plasma	↑↑↑ HMGB1 levels in ICU group compared to outpatients, and SARS-CoV-2 patients compared to healthy controls	ND	[120]
**RSV**	Bronchiolitis	**In vitro:** Primary hAECs cells	-	RSV-induced HMGB1 release (6 hpi), necroptosis is mediated via a HMGB1/RAGE	ND	[121]
**HSV-2**	Epithelium	**In vitro:** HEC-1 cells		↓↓ HMGB1 transcription. HMGB1 gradually moved from the cytosol, is transiently concentrated in the nucleus	ND	[122]
**WNV**	CNS	**In vitro:** Vero cells		Infection with high MOIs (>10) induced necrosis and the passive release of HMGB-1	ND	[123]
		**In vivo:** 29 do C57/BL6 female mice were infected with WNV (10^5 TCID50).	Brain/Serum	↑↑ HMGB1 at the early and late times p.i that potentialy caused ED. Rescue assay with neurtralizing antibodies for HMGB-1 prevent ED and supressed the expresion of TNF-α.	Yes	[124]
		**C.C:** 49 WNV-infected patients and 30 healthy controls	Serum	↑↑↑ HMGB1 levels in WNND patients than those diagnosed for WNF	Yes, for CNS alteration	[99]
**JEV**	CNS	**In vitro:** Huh7 cells		JEV inhibited the expression of HMGB1. Conversely, ↑↑ HMGB1 restricted JEV replication.	ND	[125]
		**In vitro:** HBMECs/bEnd.3 cells		**mRNA and prot:** ↑↑↑ HMGB1 secreted HMGB1 promoted the adhesion of immune cells to the EC and facilitated transendothelial migration of JEV-infected monocytes	ND	[126]
**ZIKV**	CNS	**In vitro:** Acute peripheral monoblastic leukemia-derived (MM-1) cells and HBMECs cells		↑↑ HMGB1 release that affected the integrity of HBMECs	ND	[127]
		**In vitro:** Huh7 cells		Infection induced the translocation of HMGB1 from the nucleus to the cytoplasm following its release to the extracellular environment. Dexamethasone inhibited HMGB1 mobilization and ZIKV replication only when HMGB1 was not inhibited.	ND	[128]

Human immunodeficiency virus (HIV); coxsackievirus B3 (CVB3); human T-lymphotropic virus type 1 (HTLV-1); rotavirus (RV); Rhesus rotavirus (RRV); rotavirus strain, WT-RRV; RRV mutants (RRVVP4-K187R, RRVVP4-D308A); hepatitis B virus (HBV); hepatitis C virus (HCV); influenza B (FLUBV); influenza A (FLUAV); respiratory syncytial virus (RSV); major human rhinovirus (HRV); Wisconsin virus (WV); herpes simplex virus type 2 (HSV-2) West Nile virus (WNV); Japanese encephalitis virus (JEV); Zika virus (ZIKV); conditions (Cond); severe immune deficiency (SID); biliary atresia (BA); acute-on-chronic liver failure (ACLF); hepatitis B-related Child–Pugh A cirrhosis (LC); chronic hepatitis B (CHB); hepatocellular carcinoma (HCC); portal vein tumor thrombus (PVTT); mild immune deficiency (MID); healthy controls (HC); healthy subjects (HS); central nervous system (CNS); inflammatory bowel disease (IBD); liposaccharide (LPS); long-term non-progressors (LTNP); intestinal fatty acid binding protein (I-FABP); myeloid differentiation factor 2 (MD2); HIV-associated neurocognitive disorders (HAND); cerebrospinal fluid (CSF); combined antiretroviral therapy (c-ART); days post-infection (dpi); simian immunodeficiency virus (SIV); C-reactive protein (CRP); triglycerides (Trig); glucose (Glu); cholesterol (Cho); soluble CD14 (Scd14); macaque (Mq); human (Hm); messenger RNA (mRNA); peripheral blood mononuclear cells (PBMC); hepatitis B virus X protein (HBx); BALB/nude mice (nu/nu); years-old (yo), months-old (mo), weeks-old (wo); days-old (do); intensive care unit (ICU); high elevation (↑↑↑); medium elevation (↑↑); minor elevation (↑); equal levels (↔); downregulation (↓); West Nile fever (WNF); WNV with neurological disease (WNND); ND: not described; different (≠); post-infection (p.i); endothelial damage (ED), area under the curve (AUC); and macrophages (Mø).

## Data Availability

Not applicable.

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
