# Peer review of "Endothelial Dysfunction, HMGB1, and Dengue: An Enigma to Solve"

_viruses, 2022, doi:10.3390/v14081765_

Round 1

Reviewer 1 Report

This is a well-written review on the pathogenesis of dengue virus infection, focusing on the dysfunction of endothelial cells. A major asset is the perspective summarizing the mechanism of intracellular molecular mobilization. I have a few comments below.

Fig. 1A: It is unclear why antibodies that was induced by the secondary heterotypic infection only could activate complement system. Should antibodies induced by the primary infection also activate complements?

 Fig. 1B: This figure indicates the extrinsic ADE. Authors need to add the intrinsic ADE, too.

Author Response

REVIEWER 1.

This is a well-written review on the pathogenesis of dengue virus infection, focusing on the dysfunction of endothelial cells. A major asset is the perspective summarizing the mechanism of intracellular molecular mobilization. I have a few comments below.

R/ Thank you for your kind words.

Fig. 1A: It is unclear why antibodies that was induced by the secondary heterotypic infection only could activate complement system. Should antibodies induced by the primary infection also activate complements?

R/ Thank you for letting us notice this. Yes, complement activation also happens during primary infection. We have changed the figure 1A for clarity.

Fig. 1B: This figure indicates the extrinsic ADE. Authors need to add the intrinsic ADE, too.

R/. Intrinsic ADE was added to the figure 1B.

Reviewer 2 Report

Critique

This review by Calderon-Pelaez/Coronel-Ruiz et al. focuses on the current literature regarding HMGB1 in dengue virus (DENV) infection and associated endothelial dysfunction.  The authors propose that HMGB1 may be a useful biomarker for the early diagnosis of dengue and severe complications associated with DENV infection.   This is a well-written manuscript and the illustrations are clearly presented.  I believe the review of the literature will be generally helpful to the field. I have only a few comments that may improve the review.  

Comments:

1.     Introduction:  I think the numbers/reference for SARS-2 cases could be updated to the current statistics since we are almost an entire year from September 2021.

2.     The figures are clear, but enlargement would be helpful.  

3.     Because HMGB1 is associated with many viral and bacterial infections, this would be a good place to compile the current information in a table.  This would be helpful to place the role of HMGB1 in DENV infection as it might contrast or compare with what is known about its role in other infections.  I think this might help the authors make the argument for why HMGB1 (which is fairly non-specific) would be particularly useful as a biomarker for dengue.  

4.     The clinical data for detection of HMGB1 is the most essential basis for its use as a biomarker.  The work described on page 10, lines 342-360 should include details on how big of cohorts were studied in these references.  

Minor edits:

There are a few minor typos and errors that I noticed.

1.     The author list on the title page is incomplete? (Line 5)

2.     Figure 1, panel E. “Cytokine Storm” typo as “Citokine Storm”

3.     Table 1.  Proof carefully, several “e” “y” throughout in column “other results” 

4.     Table 1. Proteins section, what is “(OD 3.0)”.  Include in definitions.

Author Response

REVIEWER 2.

This review by Calderon-Pelaez/Coronel-Ruiz et al. focuses on the current literature regarding HMGB1 in dengue virus (DENV) infection and associated endothelial dysfunction. The authors propose that HMGB1 may be a useful biomarker for the early diagnosis of dengue and severe complications associated with DENV infection. This is a well-written manuscript and the illustrations are clearly presented. I believe the review of the literature will be generally helpful to the field. I have only a few comments that may improve the review.  

R/ Thank you for your kind words.

Comments:

Introduction: I think the numbers/reference for SARS-2 cases could be updated to the current statistics since we are almost an entire year from September 2021.

R/ You are right. The numbers and reference for Sars-CoV2 has been updated to July 2022.

  1. The figures are clear, but enlargement would be helpful.  

R/ Figures have been enlarged.

Because HMGB1 is associated with many viral and bacterial infections, this would be a good place to compile the current information in a table. This would be helpful to place the role of HMGB1 in DENV infection as it might contrast or compare with what is known about its role in other infections. I think this might help the authors make the argument for why HMGB1 (which is fairly non-specific) would be particularly useful as a biomarker for dengue.

R/ We understand the addressed point, so we prepared an additional table highlighting the role of HMGB1 in different viral infections (Page 16 of the uploaded manuscript). However, regarding bacterial infection we noticed that most of what is described for HMGB1 is reported during patients’ sepsis and it is not addressed to a specific type of bacteria, which dilutes the aim of the paper. Given that bacterial infections are not our field of expertise and that it would take us a lot of time to prepare that kind of table, we prefer to add the new table only regarding viral infections.

The clinical data for detection of HMGB1 is the most essential basis for its use as a biomarker. The work described on page 10, lines 342-360 should include details on how big of cohorts were studied in these references.  

R/ Agreed, we added more detailed information in this section (Page 12).

Minor edits:

There are a few minor typos and errors that I noticed. The author list on the title page is incomplete? (Line 5)

R/ Yes, the author list is complete.

Figure 1, panel E. “Cytokine Storm” typo as “Citokine Storm”

Table 1 Proof carefully, several “e” “y” throughout in column “other results” 

Table 1. Proteins section, what is “(OD 3.0)”. Include in definitions.

R/ Thank you for letting us know. We changed all of this.

Reviewer 3 Report

In this review, the authors present dengue infections and the potential mechanisms explaining the severe forms. They propose a review of the literature, which highlights a potential role for the HMGB1 protein in these severe forms. This is a good overview of the data available from 2009 until recently to support the involvement of HMGB1 in the pathophysiology of DENV infections.

The introduction part is confusing, maybe it is related to the lack of a section. Line 25 and 104, section 1. and 3., section 2 is lacking ? maybe section 2 at line 65

Table 1 :  « Spanish like » : ¿Marker of severity?

Line 216, for HMGB1, it is better to say that the presence of a disulfide bridge between cys23 and 45 is essential for it proinflammatory activity.

Line 396-409 and 410-418 :

The authors present two facts which seem contradictory, HMGB1 allows the antiviral response (resv experiments) and the inhibition of HMGB1 which decreases the viral load. The whole ambiguity is in the existence of a pool accumulating in the nucleus or transiting into the cytoplasm to be secreted for HMGB1.

As the review is based on the potential role of HMGB1 in the ED during Dengue infection, a figure 3 will be needed to summarise all the evidence suggesting a role for HMGB1 in the ED. This will also help to show the ambiguity of HMGB1 functions depending on its subcellular location on DENV infection and pathogenesis.

Author Response

REVIEWER 3

In this review, the authors present dengue infections and the potential mechanisms explaining the severe forms. They propose a review of the literature, which highlights a potential role for the HMGB1 protein in these severe forms. This is a good overview of the data available from 2009 until recently to support the involvement of HMGB1 in the pathophysiology of DENV infections.

R/ Thank you for your kind words.

The introduction part is confusing, maybe it is related to the lack of a section. Line 25 and 104, section 1. and 3., section 2 is lacking? maybe section 2 at line 65

R/. We apologize, it was an error in the numbering of the paragraphs. In the new version this was changed.

Table 1: «Spanish like»: ¿Marker of severity?

R/ In the new version, we change the heading for “¿severity biomarker?”

Line 216, for HMGB1, it is better to say that the presence of a disulfide bridge between cys23 and 45 is essential for it proinflammatory activity.

R/ In line 216 of the original manuscript we said, “Some reports have shown that the oxidation of Cys106 is associated with the inhibition of the inflammatory activity of HMGB1”, as part of the explanation of the post-traductional modifications that HMGB1 suffers to be allowed to translocate to the cytoplasm. We do not mention Cys23 or Cys45 in this line.

In line 213 we did mention Cys23 and Cys45, as part of the protein residues that have to be oxidized to allow the conformational changes that happens to HMGB1 under stress conditions favoring its cytoplasm translocation. Reading this part of the document, we do not discuss proinflammatory activity or the effect of disulfide bonds on it, so we couldn’t find where should we make the change suggested by the reviewer.

Line 396-409 and 410-418: The authors present two facts which seem contradictory, HMGB1 allows the antiviral response (resv experiments) and the inhibition of HMGB1 which decreases the viral load. The whole ambiguity is in the existence of a pool accumulating in the nucleus or transiting into the cytoplasm to be secreted for HMGB1.

R/ We corrected, expanded, and updated the data on the effect of resveratrol - HMGB1 and dengue infection (lines 420-444)

As the review is based on the potential role of HMGB1 in the ED during Dengue infection, a figure 3 will be needed to summarize all the evidence suggesting a role for HMGB1 in the ED. This will also help to show the ambiguity of HMGB1 functions depending on its subcellular location on DENV infection and pathogenesis.

R/ A figure was added that summarizes the possible participation of the protein during the ED induced by DENV (page 14).